# Double mimicry evades tRNA synthetase editing by toxic vegetable-sourced non-proteinogenic amino acid

Youngzee Song[1], Huihao Zhou[1,5], My-Nuong Vo[1], Yi Shi[1], Mir Hussain Nawaz[1,6], Oscar Vargas-Rodriguez[2], Jolene K. Diedrich[3], John R. Yates III [3], Shuji Kishi[4], Karin Musier-Forsyth[2] & Paul Schimmel[1,4]

Hundreds of non-proteinogenic (np) amino acids (AA) are found in plants and can in principle enter human protein synthesis through foods. While aminoacyl-tRNA synthetase (AARS) editing potentially provides a mechanism to reject np AAs, some have pathological associations. Co-crystal structures show that vegetable-sourced azetidine-2-carboxylic acid (Aze), a dual mimic of proline and alanine, is activated by both human prolyl- and alanyl-tRNA synthetases. However, it inserts into proteins as proline, with toxic consequences in vivo. Thus, dual mimicry increases odds for mistranslation through evasion of one but not both tRNA synthetase editing systems.

[1] The Scripps Laboratories for tRNA Synthetase Research and the Department of Molecular Medicine, The Skaggs Institute for Chemical Biology, The Scripps Research Institute, 92037 La Jolla, CA, USA. [2] Department of Chemistry and Biochemistry, Center for RNA Biology, The Ohio State University, Columbus, OH 43220, USA. [3] Department of Molecular Medicine, The Scripps Research Institute, La Jolla, CA 92037, USA. [4] The Scripps Laboratories for tRNA Synthetase Research and Department of Molecular Medicine, The Scripps Research Institute, Jupiter, FL 33458, USA. [5] Present address: Research Center for Drug Discovery, School of Pharmaceutical Sciences, Sun Yat-sen University, 510006 Guangzhou, China. [6] Present address: Department of Chemistry, New York University, Abu Dhabi, PO Box 129188, United Arab Emirates. Correspondence and requests for materials should be addressed to P.S. (email: schimmel@scripps.edu)

The incorporation of amino acids into proteins is initiated by the 20 aminoacyl tRNA synthetases (AARSs)[1, 2]. These enzymes attach each amino acid to its cognate tRNA, which is transferred to the ribosome to be used as building blocks for protein synthesis. Similarities between some standard amino acids are so great that occasional errors in aminoacylation occur and generate mistranslated proteins. To attenuate these mistakes, synthetases evolved an editing activity that either removes mis-activated amino acids or hydrolytically clears away the mis-attached amino acid from a mischarged tRNA. In this way, the accuracy of protein synthesis is tightly controlled[3, 4].

Non-proteinogenic (np) amino acids in the food chain present yet another challenge. Many have structures similar to the 20 standard amino acids and have the potential to be mis-incorporated into proteins. Little is known about how and if AARSs have evolved mechanisms to limit the incorporation of np amino acids into cellular proteins. Azetidine-2-carboxylic acid (Aze) is a np amino acid found in sugar beets and lilies[5, 6] (Fig. 1a). Its consumption by gestating mothers is proposed to be connected with some forms of multiple sclerosis in direct off-spring[7, 8]. Based on its molecular and structural similarity to alanine (Ala) and proline (Pro) (Aze has a 4- and Pro a 5-member ring (Fig. 1a)), Aze is a dual mimic that might fit into the active site pocket of alanyl-tRNA synthetase (AlaRS) or prolyl-tRNA synthetase (ProRS) and be activated for protein synthesis. Indeed, earlier studies support the ability of Aze to be incorporated into proteins[9, 10].

Here we show that although Aze is activated by both human (Hs) AlaRS and Hs ProRS, it is rejected (>99%) by the AlaRS but not the ProRS editing system and therefore almost exclusively misincorporates into Pro positions of proteins.

## Results

**Aze fits in the active site of both Hs AlaRS and ProRS.** To establish whether Hs AlaRS and ProRS confuse Aze for Ala or Pro or both, we co-crystalized a stable analog of the aminoacyl adenylate of Aze (5′-O-[N-(L-azetidinyl) sulfamoyl] adenosine, Aze-SA) with both synthetases. The co-crystal structure (2.1 Å resolution) of Hs AlaRS with Aze-SA (Supplementary Table 1) was compared with our previous structure with Ala-SA bound to Hs AlaRS (PDB code: 4XEM, 1.28 Å). Significantly, Aze-SA could be superimposed on Ala-SA in the active site pocket of Hs AlaRS (Fig. 1b, c), and interact with the same residues that interact with Ala-SA. Thus, Aze closely mimics Ala in the active site of Hs AlaRS.

Next, we obtained a 2.6 Å co-crystal structure of Aze-SA with Hs ProRS (Supplementary Table 1), which we compared with the previously solved structure of Pro-SA with Thermus thermophilus (Tt) ProRS (PDB code: 1H4S)[11]. Aze-SA fits into the active site of Hs ProRS in almost the same conformation as Pro-SA bound to Tt ProRS (Fig. 1d, e), showing that activated Aze is also a close mimic of Pro in the active site of Hs ProRS. The dual mimicry of Aze, and its ability to 'confuse' two human tRNA synthetases, is established by these crystal structures.

**Aze toxicity is only rescued by Pro in vivo.** In order to investigate the toxic effects of Aze, we administered Aze to HeLa cell cultures and to zebrafish embryos and measured cell death after 24 h. Indeed, increased Aze concentrations in the cell media led to progressive cell death of HeLa cells (measured by trypan blue staining, Fig. 2a and Supplementary Fig. 1a), and 1 mM Aze injection into zebrafish embryos induced localized cell death detectable with increased fluorescence intensity (stained with acridine orange) in the injected embryos, when compared to uninjected controls (Fig. 2b). Because Aze is similar in structure

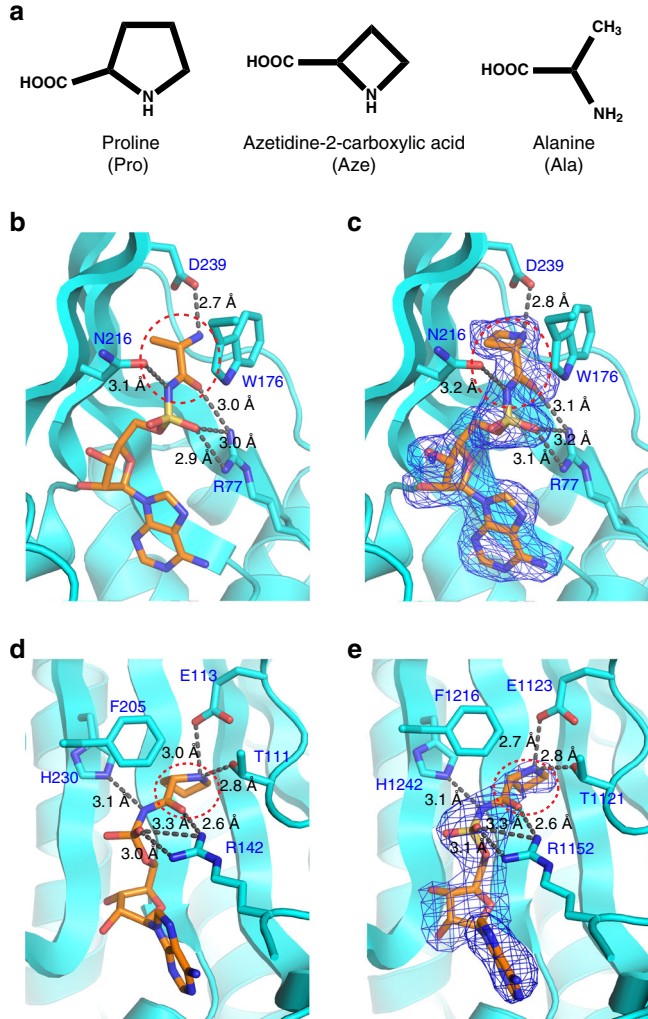

**Fig. 1** Aze fits in the active site of both Hs AlaRS and ProRS. **a** Structures of proline (Pro), azetidine-2-carboxylic acid (Aze), and alanine (Ala). Structure of Hs AlaRS with **b** Ala-SA or **c** Aze-SA in the active site. **d** Tt ProRS structure with Pro-SA in the active site. **e** Hs ProRS structure with Aze-SA in the active site

to both Pro and Ala (Fig. 1a), we separately added each amino acid together with Aze to check whether they can compete with Aze and rescue cell death. Toxicity in HeLa cells from 5 mM Aze was rescued by addition of 1 mM Pro, while 1 mM Ala, valine (Val), and threonine (Thr) did not rescue (Fig. 2c, d, Supplementary Fig. 1b, c). Similarly, we injected 1 mM Aze into zebrafish embryos with either 0.25 mM Pro, Val, Thr, or Ala. As seen with mammalian cells, Pro rescued Aze toxicity, while Ala, Val, and Thr did not (Fig. 2e). Further, in HeLa cells, when we raised Aze to 40 mM, 5 mM Pro provided substantial rescue, while 5 mM Ala did not. When 5 mM Ala was added in addition to 5 mM Pro, a small additional protective effect was observed (Supplementary Fig. 1d).

**Aze misincorporates into Pro positions.** Next, we tested whether Aze misincorporated into mammalian cell proteins, specifically at codons for Pro or Ala. We transfected HEK 293 cells with a plasmid expressing maltose binding protein (MBP). The protein was expressed (in the presence or absence of Aze in the media), purified, and subjected to mass spectrometry (Fig. 3a). In the

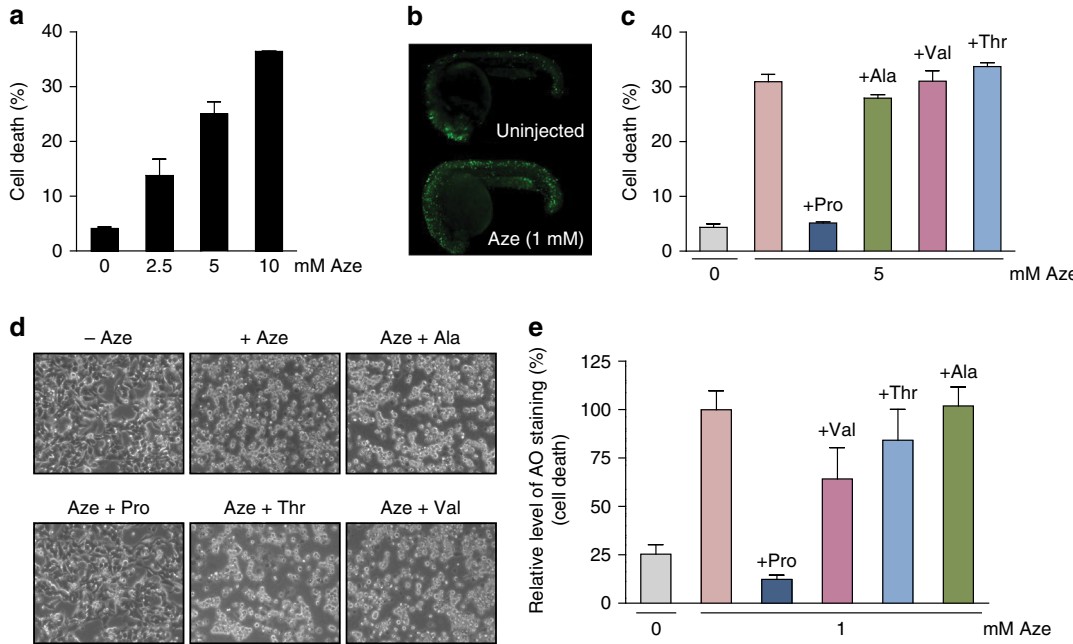

**Fig. 2** Pro rescues Aze toxicity in mammalian cells and zebrafish. **a** Cell death measured after 24 h of Aze treatment on HeLa cells. **b** Acridine orange (AO) staining of 24 h.p.f. zebrafish uninjected or injected with 1 mM Aze visualized under a fluorsecent microscope. **c** HeLa cells were treated with 5 mM Aze with either no amino acid, 1 mM Ala, Pro, Val, or Thr. Cell death was quantified and plotted. **d** Image of cells for **c**. **e** Quantification of AO staining of 24 h.p.f zebrafish uninjected, or injected with 1 mM Aze, or 1 mM Aze with either 0.25 mM Pro, Val, Thr, or Ala. (Error bars represent s.d. (**a**, **c**) or s.e.m. (**e**))

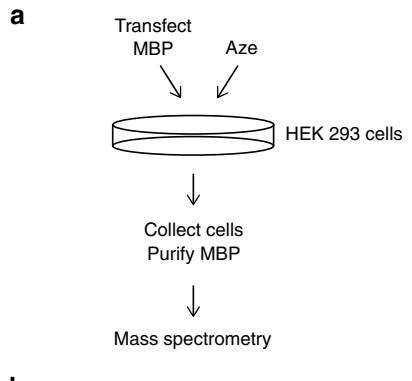

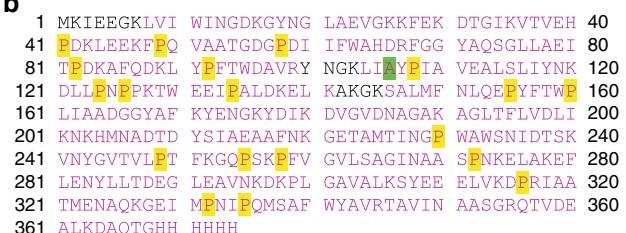

**Fig. 3** Aze misincorporates into Pro positions of proteins. **a** Experimental procedure for detection of Aze misincorporation into proteins. **b** MBP sequence; 96% coverage by mass spectrometry analysis (sequence covered colored in magenta). Pro positions replaced by Aze highlighted in yellow, and the single Ala position replaced by Aze at a frequency of 0.024% highlighted in green

| Table 1 Aze detection in mammalian cell purified MBP at Pro positions | | | | |
|---|---|---|---|---|
| | +Aze sample | | −Aze sample | |
| Pro position | Aze detected | Pro detected | Aze detected | Pro detected |
| 41 | 46 | 182 | 0 | 430 |
| 49 | 33 | 206 | 0 | 486 |
| 58 | 15 | 221 | 0 | 472 |
| 82 | 43 | 459 | 0 | 673 |
| 92 | 9 | 35 | 0 | 77 |
| 108 | 14 | 58 | 0 | 244 |
| 124 | 3 | 44 | 0 | 83 |
| 126 | 3 | 44 | 0 | 83 |
| 127 | 0 | 47 | 0 | 83 |
| 134 | 8 | 56 | 0 | 71 |
| 155 | 4 | 12 | 0 | 21 |
| 160 | 3 | 13 | 0 | 21 |
| 230 | 7 | 27 | 0 | 56 |
| 249 | 11 | 21 | 0 | 28 |
| 255 | 6 | 119 | 0 | 144 |
| 258 | 35 | 112 | 0 | 157 |
| 272 | 22 | 125 | 0 | 157 |
| 299 | 0 | 7 | 0 | 27 |
| 316 | 5 | 19 | 0 | 38 |
| 332 | 18 | 48 | 0 | 55 |
| 335 | 20 | 46 | 0 | 55 |
| Total | 305 | 1901 | 0 | 3461 |

ensemble of statistical proteins that were generated, 21 Pro positions were collectively detected 2206 times, and 305 of these detections contained Aze (13.8%, Fig. 3b). No Aze was detected in the non-Aze-treated samples (Tables 1 and 2, Supplementary Fig. 2). Aze was also found at one Ala position at a frequency of

0.024%. For the Pro → Aze replacements, the substitutions were randomly scattered across the sequence. However, we noted that some positions were more susceptible to replacement than others (Table 1). For example, Pro 255 was replaced by Aze at a frequency of 6/125, while Pro 258 was replaced by Aze at a ratio of 35/147. Possibly, Aze fits into some places in the sequence better

than others, with the poorer fits resulting in protein instability and destruction.

**Aze is activated by both *Hs* AlaRS and ProRS.** Given the clear dual Ala, Pro mimicry of Aze, the above observations raised the question of why Aze was incorporated specifically at positions for Pro, and not also for Ala. For this purpose, we studied the activation of Aze by both enzymes. For all the experiments with *Hs* ProRS, recombinant protein corresponding to only the ProRS portion of the natural human glutamyl–prolyl–tRNA synthetase (GluProRS) fusion protein was used. Consistent with observations of the crystal structures, both enzymes activated Aze (Fig. 4a–c; no contamination with other amino acids was detected above 0.01% in our Aze stock.) However, at the same concentration of amino acid (1 mM), activation of Aze by *Hs* ProRS (catalytic efficiency ($k_{cat} K_m^{-1}$)) was 71-fold slower than with Pro (Supplementary Table 2), and much more robust than with Ala, an amino acid known to be misactivated by ProRS[12] (Fig. 4a). For *Hs* AlaRS, the efficiency of activation of Aze was similar to its misactivation of serine (Ser) (Fig. 4c, and higher than the negative controls using Thr and Pro), which is known to be cleared away by editing[13]. In particular, Aze and Ser had 1229- and 950-fold

---

**Table 2 Summary of Aze misincorporation into mammalian cell purified MBP, in Pro and Ala positions**

|  | +Aze sample | −Aze sample |
|---|---|---|
| Pro → Pro | 1901 | 3461 |
| Pro → Aze | 305 | 0 |
| Ala → Ala | 4108 | 7842 |
| Ala → Aze | 1 | 0 |

13.8% of 21 Pro positions detected were replaced with Aze, 0.024% of 43 Ala positions detected were replaced with Aze

---

lower relative catalytic efficiency ($k_{cat} K_m^{-1}$), respectively, compared to Ala (Supplementary Table 2).

**Aze is edited by *Hs* AlaRS but not by *Hs* ProRS.** Two types of editing prevent mistranslation. One is pre-transfer editing, where the misactivated aminoacyl adenylate is hydrolyzed to prevent the transfer of the amino acid to tRNA[14, 15] (proposed to occur within the synthetic active site, distinct editing active site, or after release into solution[16]). Another is post-transfer editing, where the mischarged amino acid is hydrolyzed to prevent its misincorporation into proteins[4]. *Hs* AlaRS showed pre-transfer activity against both Ser and Aze ($0.09\,s^{-1}$), with no activity against Ala (as expected) (Fig. 5a).

AlaRS has an editing domain that hydrolyzes mischarged Ser from Ser-tRNA$^{Ala}$[13]. We used a truncated version of *Hs* AlaRS (AlaRS$^{AD}$) in which only the aminoacylation domain (AD) was retained after ablation of the editing domain (Fig. 5b). We compared its activity to that of full-length (FL) *Hs* AlaRS (AlaRS$^{FL}$). Both AlaRS$^{FL}$ and AlaRS$^{AD}$ efficiently charged tRNA$^{Ala}$ with Ala, but Ser was only charged by AlaRS$^{AD}$ (Fig. 5b). In contrast to Ser, Aze was not efficiently charged by either AlaRS$^{FL}$ or AlaRS$^{AD}$. Thus, pre-transfer editing activity is effective in removing misactivated Aze-AMP.

While *Hs* ProRS lacks a dedicated editing domain, such as found in most bacterial ProRSs, editing of mischarged tRNA$^{Pro}$ in human cells may be achieved by a separate genome-encoded editing factor, ProXp-ala. This factor clears Ala-tRNA$^{Pro}$ by hydrolytic editing[17]. To test whether *Hs* ProXp-ala editing activity acts on Aze, purified *Hs* ProXp-ala was incubated with tRNA$^{Pro}$ mischarged either with Ala or Aze. While *Hs* ProXp-ala hydrolyzes Ala-tRNA$^{Pro}$, it is unable to clear Aze-tRNA$^{Pro}$ (Fig. 5c). Therefore, *Hs* ProXp-ala does not have editing activities against Aze.

In a competition assay, we observed that Aze could efficiently compete with Pro for charging onto tRNA$^{Pro}$ by *Hs* ProRS

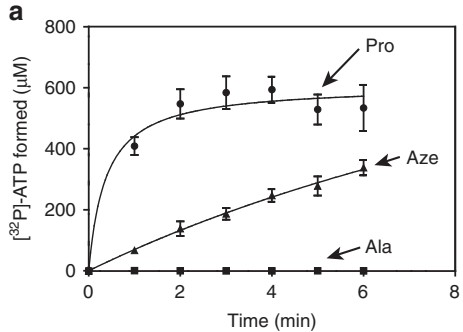

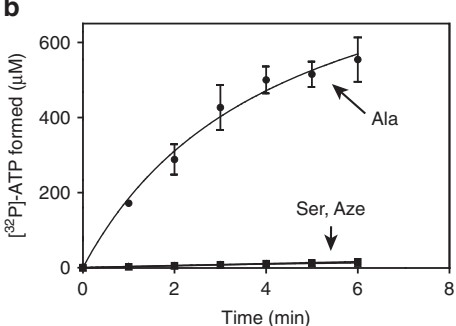

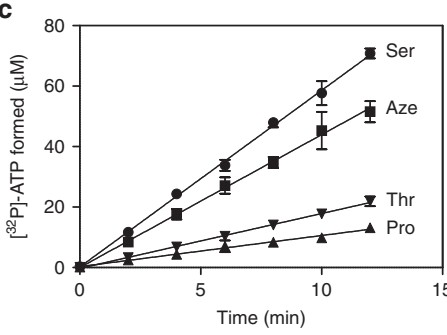

**Fig. 4** Activation of Aze by *Hs* ProRS and *Hs* AlaRS. **a** Formation of [$^{32}$P]-ATP by 0.25 µM *Hs* ProRS with either 1 mM Pro, Ala, or Aze. **b** Formation of [$^{32}$P]-ATP by 0.25 µM *Hs* AlaRS with either 1 mM Ala, Ser, or Aze. **c** Formation of [$^{32}$P]-ATP by 0.5 µM *Hs* AlaRS with either 1 mM Ser, Aze, Thr, or Pro. (Error bars represent s.d.)

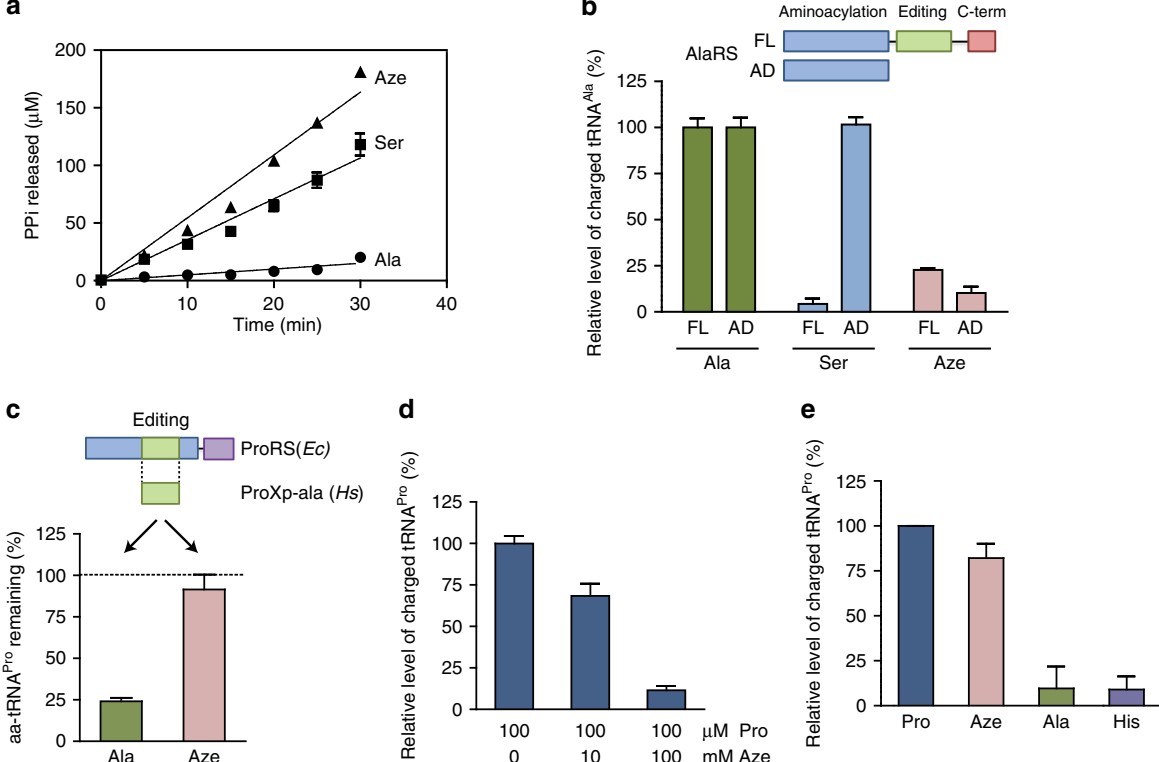

**Fig. 5** *Hs* AlaRS has pre-transfer editing activity against Aze, *Hs* ProRS has no editing activity against Aze. **a** ATP hydrolysis in the presence of 20 mM Ala, 100 mM Ser, or 100 mM Aze with 1 µM *Hs* AlaRS. **b** Relative level of charged tRNA^Ala after aminoacylation for 5 min by full-length *Hs* AlaRS (FL) or aminoacylation domain of AlaRS (AD) with either 2.5 mM Ala, Ser, or Aze. **c** Deacylation assay with *Hs* ProXp-ala using either Ala-tRNA^Pro or Aze-tRNA^Pro. **d** Aminoacylation of tRNA^Pro by *Hs* ProRS with Pro in the presence of different concentrations of Aze at 20 min. **e** Relative level of charged tRNA^Pro after 20 min aminoacylation by *Hs* ProRS with either 2.5 mM Pro, Aze, Ala, or His. (All graphs represent mean ± s.d., average of three experiments.)

(Fig. 5d, Supplementary Fig. 3a). In contrast, Ser did not compete with Pro (Supplementary Fig. 3b). Activation of Aze also resulted in significant production of Aze-tRNA^Pro (Fig. 5e, Supplementary Fig. 3c). Consistent with an earlier report, Aze clearly escaped the editing function of *Hs* ProRS[18].

## Discussion

More than 50 years ago, hundreds of non-protein amino acids were identified in plants[5]. Due to its 4-member ring structure, Aze has long been recognized as a mimic of Pro. Our work confirmed earlier studies that showed, mostly by indirect methods, Aze can replace Pro in proteins[10, 19]. Importantly, our co-crystal of *Hs* ProRS with the adenylate analog of Aze-AMP was closely similar to that of *Tt* ProRS in complex with the Pro-AMP analog. However, the close similarity of the Aze side chain to the methyl group of Ala raised the question of why Ala also would not be susceptible to replacement by Aze. Our thought was that the active site of AlaRS might discriminate against Aze and reject it from being activated with ATP to form Aze-AMP. In this instance, we thought that a complex of AlaRS with Aze-AMP would either not occur or would be clearly distinct from that of AlaRS with Ala-AMP. Surprisingly, our co-crystal structure revealed a complex of *Hs* AlaRS with the Aze-AMP analog. This complex was closely similar at the atomic level to that of the corresponding complex with the cognate Ala-AMP analog. This robust fit of Aze-AMP into both active sites led us to speculate that Aze could be used by both tRNA synthetases and charged onto their respective tRNAs.

Surprisingly, while both *Hs* ProRS and *Hs* AlaRS activated Aze, only ProRS was successful in completing the charging event. On more closely inspecting the crystal structures, we could not visualize any differences, such as catalytic water molecules, which would make the AlaRS complex more labile and therefore subject to breakdown of the adenylate. While this may reflect a need for an even higher resolution of the structures to eliminate the possibility of an active water molecule, the data we obtained clearly pointed to the capacity of AlaRS, and not ProRS, to use its pre-transfer editing function to clear away activated Aze.

The widespread distribution of Aze at positions for Pro (observed in our mass spectrometry analysis of MBP) gives rise to an immense population of statistical proteins. Some of the micro species in this statistical population are undoubtedly antigenic. As highlighted by Rubenstein[8], newborn lambs born from mothers fed with Aze-containing sugar beet silage were shown to have an autoimmune multiple-sclerosis-like phenotype[7]. Thus, while a causal link has not been established, this observation leads to speculation that Aze may be a contributing factor. This contribution might be especially penetrant at the vulnerable embryonic development stage, where mistranslated proteins may have more severe effects. Here we found that Aze is toxic to zebrafish embryos and that toxicity was rescued by administration of Pro. While our experiments used high concentrations of Aze, a reduced range of Aze dosage could lead to low levels of mistranslated proteins, with the potential to cause autoimmunity[8].

Our results show that while dual mimic Aze was captured for removal by one tRNA synthetase (AlaRS), it evaded the other (ProRS[18] and ProXp-Ala). Additional examples of multiple

mimicry include, among others, beta-N-methylamino alanine (BMAA), canaline, and ethioine (Supplementary Table 3, showing all 7). Thus, the problem for preventing a dual mimic's Trojan Horse-like entry into proteins is more general than what is presented here for Aze, and may be reflected in disease associations such as those correlated with Aze[7, 8] and BMAA[20–22] consumption.

## Methods

**Cell death induced by Aze in mammalian cells.** HeLa cells (ATCC, mycoplasma tested) were cultured in 10% FBS, DMEM media (Gibco). To test Aze toxicity, HeLa cells were plated in 2% FBS DMEM media. After the cells attached to the plate, Aze (Sigma-Aldrich or Santa Cruz) was added to the cell medium at different concentrations; 24 h later, cell death was measured by Beckman-Coulter ViCell XR. To test whether adding different amino acids can rescue Aze toxicity, experiments were performed as mentioned above, except cells were treated with 5 mM Aze or 5 mM Aze with either 1 mM Ala, Pro, Val, or Thr. To determine the amount of Pro needed to rescue Aze toxicity, HeLa cells were treated with 5 mM Aze and either 1000 μM, 500 μM, 100 μM, 50 μM, 10 μM Pro or no Pro. Cell death was measured 24 h later. Bar graphs show average of two independent experiments. For Supplementary Fig. 1d, HeLa cells were treated with 40 mM Aze, with either 5 mM Pro or Ala, or Pro and Ala (in 2% FBS DMEM); 24 h later, viable cell numbers were quantified using Cell Titer 96 AQueous Non-Radioactive Cell Proliferation assay kit (Promega). The data represent two individual experiments performed in triplicates.

**Aze induced cell death in zebrafish embryos.** Zebrafish were maintained at 28.5 °C under continuous water flow and filtration with automatic control for a 14-h/10-h light/dark cycle[23]. Single-cell stage embryos were injected with either 1 mM Aze, or 1 mM Aze with 0.25 mM Pro, Val, or Thr ($n = 11$–40, no randomization or blinding). At 24 h.p.f., zebrafish embryos were AO stained. Live zebrafish embryos were dechorionated in pronase (2.0 mg ml$^{-1}$ in egg water, 5 mM NaCl, 0.17 mM KCl, 0.33 mM CaCl$_2$, 0.33 mM magnesium sulfate) and rinsed with egg water. Embryos were then incubated in 10 μg ml$^{-1}$ AO (Sigma A-6014) in egg water for 15 min, followed by five quick rinses[23] and photographed with a Nikon fluorescent microscope (AZ100) equipped with a Nikon CCD camera (QimagingRetiga 2000R). Error bars in the graph represent standard error of the mean (s.e.m). The zebrafish experiments were conducted according to the guidelines established by the Institutional Animal Care and Use Committee (IACUC) at The Scripps Research Institute.

**Protein purification.** pET21a (Novagen) plasmid containing either his-tagged human ProRS (only the ProRS portion of the fusion protein GluProRS) or pET28a (Novagen) AlaRS (FL and AD (1–455aa)) was transformed into *Escherichia coli* BL21 (DE3) cells. Expression of proteins was induced with 0.2–0.5 mM isopropyl β-D-1-thiogalactopyranoside (IPTG) at room temperature overnight. Overexpressed proteins were purified from *E. coli* with Ni-NTA resin (Qiagen), and buffer exchanged in either 50 mM HEPES pH 7.5, 150 mM NaCl buffer, or PBS buffer. Proteins were concentrated and stored at −80 °C. Concentrations of proteins were determined by active site titration[24].

Plasmid pT7T3D-PacI harboring the human ProXp-ala cDNA was purchased from Open Biosystems (ThermoFisher). According to GenBank accession number A6NEY8, this construct lacks portions of the 5′- (18 bp) and 3′-ends (30 bp) of the gene *proXp-ala* gene. Thus, the corresponding bases were incorporated into the gene by three consecutive rounds of PCR amplification (*Pfu* DNA polymerase) using three distinct forward primers (5′-GCCATCCACACTGAGGTCGTGGAGCACCCGAGCTATTTACAGTTGAAGAAATG-3′, 5′-GCGCTCGAGCAGCGGCTCGGTGCCCTGGCCATCCACACTGAGGTCGTG-3′, and 5′-GGGAATTCCATATGCGGGCGCGGAGTTGGGAGCGGCGCTGGAGCAGCGG-3′) and one single reserve primer (5′-CGCGGATCCGCGTTAGTTGTTTTTATCAAAATTTAGTATTATGGGATCATGTCC-3′). Forward and reverse primers contained flanking restriction sites (underlined) for NdeI and BamHI endonucleases (NEB), respectively. The gene was cloned into pET15b (Novagen). Human ProXp-ala was overexpressed in *E. coli* BL21 (DE3) cells overnight at room temperature after addition of 0.1 mM IPTG. The his-tagged protein was purified using a HIS-select® nickel resin (Sigma-Aldrich), and eluted with an imidazole gradient containing 300 mM NaCl, 50 mM NaH$_2$PO$_4$ pH 8.0, and 2 mM β-ME. Purified ProXp-ala was stored in 50 mM Tris-HCl pH 7.5, 100 mM NaCl, and 2 mM DTT buffer. ProXp-ala protein concentrations were estimated by the Bradford assay (BioRad)[25].

**Crystallization.** AlaRS AD was digested with trypsin (to improve its crystallizability) overnight at 20 °C, and further purified with a monoQ 10/100 GL column (GE Healthcare). The purified AlaRS was concentrated and stored in 2 mM Tris-HCl pH 8.0, 50 mM NaCl buffer at −80 °C.

Prior to crystallization, ProRS (40 mg ml$^{-1}$) and AlaRS (30 mg ml$^{-1}$) were both incubated with 2 mM 5′-O-(N-(L-azetidine-2-carbonyl)-sulfamoyl) adenosine (Aze-SA) on ice for 30 min. Crystallization was performed by sitting-drop vapor-diffusion method by mixing 100 nl protein and 100 nl reservoir solution, and then equilibrated against 60 μl reservoir solution for 2–3 days. Large ProRS crystals were obtained at 20 °C with a reservoir solution of 9% PEG 3350, 250 mM NaNO$_3$, and 100 mM 3-(1-Pyridino)-1-propane sulfonate. Large AlaRS crystals grew at 37 °C under a reservoir solution of 25% PEG 6000 and 100 mM Tris-HCl pH 8.0.

Diffraction data were collected with a single flash-cooled crystal at 100 K on an in-house Cu Kα X-ray system equipped with the Mar345 image plate (MAR research, Germany), and processed with HKL2000. ProRS–Aze-SA and AlaRS–Aze-SA complex structures were solved with molecular replacement using previously reported human ProRS structure (PDB code: 4HVC) and AlaRS structure (PDB code: 4XEM) as templates. The final models was refined to 2.59 Å with Rwork = 22.7% and Rfree = 27.2% for ProRS and 2.03 Å with Rwork = 18.0% and Rfree = 21.5%, and their stereochemical quality was validated with MolProbity.

**ATP-PPi exchange assay.** The ATP-PPi exchange assay was carried out in the following buffer: 100 mM Tris-HCl, pH 8.0, 10 mM MgCl$_2$, 2 mM ATP, 2 mM DTT, 10 mM KF, 0.2 mg ml$^{-1}$ BSA, 1 mM NaPPi, and [$^{32}$P]-NaPPi, and reaction was initiated with the addition of 0.05 μM *Hs* ProRS or 0.1 μM *Hs* AlaRS (carried out at 37 °C)[26]. For the assays with ProRS, Pro concentrations ranged from 0.03 to 2 mM. Aze concentrations ranged from 1 to 30 mM. For the assays with *Hs* AlaRS, Ala concentrations ranged from 0.03 to 2 mM, Aze concentrations ranged from 2 to 300 mM. The reactions were quenched in quench solution (1 M HCl, 200 mM sodium pyrophosphate, 4% charcoal) at specific time points. Kinetic parameters were determined by fitting data (from three independent experiments) to the Michaelis–Menton equation with GraphPad Prism (± standard error). To compare the [$^{32}$P]-ATP formation of different amino acids, 1 mM Ala, Pro, Ser, Thr, or Aze was used with either 0.25 μM *Hs* ProRS or *Hs* AlaRS (0.5 μM *Hs* AlaRS was used for Fig. 4c). The graph shows the average values from two independent experiments (carried out in duplicates).

**ATP hydrolysis assay.** ATP hydrolysis assay was set up as follows: 100 mM HEPES pH 7.5, 20 mM KCl, 10 mM MgCl$_2$, 1 mM ATP, 2 U ml$^{-1}$ inorganic pyrophosphatase, 2 mM DTT, and γ-[$^{32}$P]-ATP with either 20 mM Ala, 100 mM Ser, or 100 mM Aze. Reactions were initiated by the addition of 1 μM *Hs* AlaRS, and aliquots of the reaction were quenched in 1.4% HClO$_4$, 8% charcoal, 10 mM NaPPi buffer at specific time points. Plate format was used[26], and reactions were carried out at room temperature.

**TLC-based aminoacylation assays.** Human tRNA$^{Pro}$ was prepared with the in vitro run off transcription method[27]. Briefly, pUC57 vector containing human tRNA$^{Pro}$ was purified from *E. coli* with a Giga-prep kit (Qiagen). The purified plasmid was digested with BstNI, and used as a template in a T7 polymerase transcription reaction. The transcribed tRNA$^{Pro}$ was purified in a Mono Q column, and refolded by heating to 80 °C and adding 2 mM MgCl$_2$ while cooling. Human tRNA$^{Ala}$ was prepared using a DNA template containing a T7 promoter and human tRNA$^{Ala}$ gene that was synthesized by PCR of overlapping oligonucleotides. The transcription reaction was performed using MEGAshortscript T7 kit (ThermoFisher) according to the manufacturer's instructions. The DNA template was then digested with DNase and the tRNA transcript was purified by phenol–chloroform extraction, ethanol precipitated and refolded.

tRNA was [3′-$^{32}$P] labeled by incubating tRNA with 50 mM sodium pyrophosphate, 0.2 μM tRNA nucleotidyltransferase, 0.3 μM [α-$^{32}$P] ATP, 10 mM MgCl$_2$, and 50 mM glycine (pH 9.0) at 37 °C for 5 min followed by addition of 1 μM CTP and 10 U ml$^{-1}$ PPiase and an additional 2 min incubation[28]. For *Hs* AlaRS, aminoacylation reactions were carried out at room temperature in a mixture of 500 nM human AlaRS$^{FL}$ or AlaRS$^{AD}$ with 20 mM amino acids (Ala, Ser, or Aze), and 1 μM [3′-$^{32}$P] labeled tRNA$^{Ala}$. For *Hs* ProRS, the reaction was performed with 100 nM ProRS at room temperature in the presence of 2.3 μM [3′-$^{32}$P] labeled tRNA$^{Pro}$, and 2.5 mM amino acids (Pro, Aze, Ala, or histidine (His)). Aliquots of the reaction were quenched with sodium acetate (pH 5.0) followed by phenol–chloroform extraction, ethanol precipitation, and digestion with either P1 nuclease (Sigma-Aldrich) or S1 nuclease (ThermoFisher). After digestion, aliquots were spotted on PEI cellulose plates (EMD Millipore), and run in glacial acetic acid/1 M NH$_4$Cl/water (5:10:85) solution. Aminoacyl-[$^{32}$P]-AMP and [$^{32}$P]-AMP were separated on a TLC plate and imaged on the phosphoimager. Percent of aminoacylation was quantified with the Molecular Dynamics Image Quant software. The level of charged tRNA was normalized to the Ala-tRNA$^{Ala}$ or Pro-tRNA$^{Pro}$ aminoacylated by either FL *Hs* AlaRS or *Hs* ProRS. Before normalization, all levels of charged tRNA were corrected by subtracting the background in the absence of amino acid with the corresponding protein. (Maximal charging levels of the tRNAs were 40–50%.)

**Aze competition with Pro for charging.** For the aminoacylation assays, tritium-labeled Pro was used with *Hs* ProRS and tRNA$^{Pro}$. The reaction contained 50 mM HEPES pH 7.5, 20 mM KCl, 4 mM ATP, 2 mM DTT, 10 mM MgCl$_2$, 4 μg ml$^{-1}$ pyrophosphatase, 2 μM tRNA$^{Pro}$, 1.3 μM [$^3$H]-Pro (PerkinElmer), and 98.7 μM

cold Pro. Increasing concentrations of Aze (or Ser as a negative control) were added (0–100 mM) to the reaction to test whether Aze could compete with Pro for charging; 100 nM ProRS was added to initiate the reaction at room temperature. Reactions were quenched in 0.5 mg ml$^{-1}$ DNA, 100 mM EDTA, 300 mM NaOAc pH 3.0 buffer, with addition of 20% TCA for tRNA precipitation (in a filter plate (Millipore)). The wells were washed in ice-cold 5% TCA, 100 mM Pro solution, dried, and tRNA was hydrolyzed by addition of 0.1 M NaOH. The plate was centrifuged, and the flow through was mixed with supermix cocktail, and the amount of [$^3$H]-Pro-tRNA$^{Pro}$ formed was quantified on the MicroBeta plate reader (PerkinElmer).

**Deacylation assays.** Human tRNA$^{Pro}$ was [3′-$^{32}$P] labeled[28]. Ala- and Aze-tRNA$^{Pro}$ were prepared by incubating 4 μM human ProRS with 8 μM tRNA and the corresponding amino acid (500 mM Ala or 500 mM Aze) in buffer containing 25 mM HEPES pH 7.0, 4 mM ATP, 25 mM MgCl$_2$, 0.1 mg ml$^{-1}$ BSA, 20 mM KCl, and 20 mM β-ME for ~2 h at room temperature.

Post-transfer editing experiments were performed at 30 °C in reactions containing 0.75–1.5 μM aa-tRNA in 50 mM HEPES pH 7.5, 0.1 mg ml$^{-1}$ BSA, 5 mM MgCl$_2$, 2 mM DTT, 20 mM KCl, and 15 μg ml$^{-1}$ inorganic pyrophosphatase buffer. Reactions were initiated by addition of 5.2 μM human ProXp-ala and 2 μl were quenched after 25 min in a solution containing ~0.5 U μl$^{-1}$ P1 nuclease (Sigma-Aldrich) in 200 mM NaOAc pH 5.0. Aminoacyl-[$^{32}$P]-AMP and [$^{32}$P]-AMP were separated on polyethyleneimine-cellulose TLC plates and analyzed as described above.

**Mammalian MBP purification.** MBP from pMAL-c4X (NEB) was cloned into a pCDNA6 vector. HEK 293 cells (ATCC, mycoplasma tested) were transfected with pCDNA6-MBP, and 4 h later media was changed to 2% FBS DMEM media, with or without 1.2 mM Aze; 47 h after transfection, cells were harvested and lysed through sonication in 20 mM Tris-HCl pH 7.5, 250 mM NaCl, 10 mM imidazole, 5 mM β-Me buffer (lysis buffer). Lysed cells were then centrifuged at 16k rpm, 15 min at 4 °C. MBP was purified from the supernatant with Ni-NTA resin (Qiagen) and eluted in lysis buffer containing 100 mM imidazole. Purified MBP was concentrated and buffer exchanged to reduce the imidazole concentration to 10 mM.

**Mass spectrometry analysis.** MBP samples were reduced with tris (2-carboxyethyl) phosphine hydrochloride (TCEP, Sigma-Aldrich) and alkylated with chloroacetamide (Sigma-Aldrich). Proteins were digested overnight at 37 °C in 2 M urea/100 mM TEAB, pH 8.5, with trypsin (Promega). The digested samples were analyzed on a Q Exactive mass spectrometer (ThermoFisher). The digest was injected directly onto a 15 cm, 100 μm ID column packed with Aqua 3 μm C18 resin (Phenomenex). Samples were separated at a flow rate of 300 nl min$^{-1}$ on an Easy nLCII (Thermo). Buffers A and B were 0.1% formic acid in water and acetonitrile, respectively. A gradient of 1–35% B over 180 min, an increase to 80% B over 40 min, and held at 80% B for a final 5 min of washing before returning to 1% B was used for 240 min total run time. Column was re-equilibrated with buffer A prior to the injection of sample. Peptides were eluted directly from the tip of the column and nanosprayed directly into the mass spectrometer by application of 2.5 kV voltage at the back of the column. The Q Exactive was operated in a data-dependent mode. Full MS[1] scans were collected in the Orbitrap at 70 K resolution with a mass range of 400–1800 m/z. The 10 most abundant ions per cycle were selected for MS/MS and dynamic exclusion was used with exclusion duration of 15 s.

Protein and peptide identification were done with Integrated Proteomics Pipeline—IP2 (Integrated Proteomics Applications). Tandem mass spectra were extracted from raw files using RawConverter[29] and searched with ProLuCID[30] against human UniProt database appended with the transfected protein sequences. The search space included all fully tryptic and half-tryptic peptide candidates. Carbamidomethylation on cysteine was considered as a static modification. Aze incorporation was considered at proline with a mass difference of −14.01565 and at alanine with a mass difference of +12.0. Data were searched with 50 ppm precursor ion tolerance and 600 ppm fragment ion tolerance and filtered to 10 ppm at the precursor level. Identified spectra were filtered using DTASelect[31] and utilizing a target-decoy database search strategy to control the false discovery rate to 1% at the spectrum level.

**Aze stock mass spectrometry analysis.** Aze stock from Sigma or Santa Cruz were analyzed on an Agilent 6495 triple quadrupole mass spectrometer in MRM mode coupled to an Agilent 1290 UPLC stack. Column was from Imtakt (Amino Acid 2.0 × 150 mm). Mobile phase A = ACN/THF/100 mM ammonium formate/formic acid 9/75/16/0.3 v:v; Mobile phase B = ACN/100 mM ammonium formate 20/80; Gradient: T = 0 100/0, T = 12 100/0, T = 15 0/100, T = 25 off; post time re-equilibration time = 5 min; Flowrate = 0.3 ml min$^{-1}$ with 5 μl injected.

**Statistical analysis.** Statistical analyses were carried out using the GraphPad Prism software. Data are presented as mean ± s.d., unless stated otherwise.

**Data availability.** The coordinates and structure factor files of the ProRS:Aze-SA complex (PDB code: 5V58) and AlaRS:Aze-SA complex (PDB code: 5V59) have been deposited in the Protein Data Bank (PDB). All relevant data are available from the corresponding authors upon reasonable request.

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

## Acknowledgements

We thank Professor Piet Herdewijn, from Catholic University of Leuven, Belgium, for the gift of Aze-SA. We thank Dr. Sanchita Hati for cloning human ProXp-ala. This work was supported by the National Foundation for Cancer Research and NIH grant GM23562 (to P.S.), NIH grant GM113656 (to K.M.-F.) and the National Institute of General Medical Sciences 8 P41 GM103533 (to J.R.Y.).

## Author contributions

Y.So., H.Z., M.N.V., Y. Sh., M.H.N., O.V.-R., and J.K.D. performed the experiments, Y. So., J.K.D., J.R.Y., S.K., K.M.-F., and P.S. analyzed the data, and Y.So. and P.S. wrote the manuscript.

## Additional information

**Competing interests:** The authors declare no competing financial interests.

