## [Peer Review File · Nature Communications]

Reviewers' comments:

Reviewer #1 (Remarks to the Author):

In this manuscript, the authors report that Aze is confused by AlaRS and ProRS for Ala or Pro both. Also it is reported that Aze is misincorporated into protein as Pro on the basis that AlaRS can edit Aze-tRNA(Ala), whereas ProRS fails to edit Aze-tRNA(Pro). This is very interesting paper and a significant body of work, which is suitable for publication in Nature Communications. But there are some concerns need to be addressed.

1. In Fig. 1, omit electron density maps of Aze are not presented.
2. In Fig. 2f, the activation of Aze by AlaRS by is too slight—slighter than the misactivation of Ser. To clearly indicate that AlaRS activates Aze, the data AlaRS does not activate some other amino acids required as a negative control. Otherwise the description “both enzymes activated Aze (page 4, line 81)” should be toned down.
3. In Fig. 3d, negative controls for competition experiments should be supplied.
4. Please mark the labels of AlaRS domains in Fig. 3b.

Reviewer #2 (Remarks to the Author):

The manuscript by Song et al demonstrates that azetidine-2-carboxylic acid, a non-proteinogenic amino acid found in some vegetables, acts as a proline and alanine analogue in the activation/aminoacylation reactions of human ProRS and AlaRS, respectively. At the same time, Aze evades the proofreading pathways that control the accuracy of proline translation, but is edited by the AlaRS pre-transfer editing activity. Misincorporation of Aze at proline sites was demonstrated in the case of human myelin basic protein expressed in HEK 293 cells. The observed mistranslation potentially explains the cellular toxicity of Aze that was demonstrated on both HeLa cells and zebrafish embryos.

The manuscript presents interesting findings regarding the toxicity of non-proteinogenic amino

acids that are able to evade AARS quality control and participate in protein synthesis. These amino acids are present in food and were proposed to exert harmful effects in humans. Understanding the mechanisms by which they act and by which cell can defend itself from their toxicity is important. However, the manuscript lacks insights into the mechanisms underlying Aze participation in the AARS synthetic and editing pathways. Thus in this form, it does not substantially contribute to novel understanding in the field of translational quality control.

In particular, the authors showed that Aze can be activated and edited at the pre-transfer editing level by Hs AlaRS. They solved the crystal structure of AlaRS:Aze-AMS, and report that it strongly resembles the structure of AlaRS bound to the cognate Ala-AMS. However, as Aze-AMP is hydrolyzed within the Hs AlaRS active site (due to pre-transfer editing of Aze), one would expect to observe some differences between these two structures. Did the authors try to find a putative catalytic water? Could they provide any structural insights into why Aze is edited in the catalytic active site of AlaRS and Ala is not? And if not, can they comment on that?

The data showed that pre-transfer editing does not contribute to the editing of serine during Ser-tRNA^{Ala} synthesis (Fig 3b; see the full aminoacylation profile with AD), while in the case of Aze it seems to be responsible for the low level of Aze-tRNA^{Ala} formation. Meanwhile, the pre-transfer editing rates for both serine and Aze are highly similar and (presumably) slow (Fig 3a, can you please provide the rates). Why do similarly slow tRNA-independent pre-transfer editing reactions contribute to the editing only in the case of Aze, and not Ser? This should be explained and further experimentally addressed by measuring the isolated transfer step (observed mischarging up to 25% indicates it may be possible) and other editing assays. The question is important as the data suggest participation of the transfer step in discrimination against Aze.

The authors showed that Aze mimics Pro in Pro-tRNA^{Pro} synthesis and escaped the editing function of Hs ProRS (Fig 3d and 3e). This is in agreement with their previous work (Beuning and Musier-Forsyth, 2001, *J. Biol. Chem.* 276, 30779–30785) where they demonstrated that Hs ProRS lacks the pre-transfer editing activity against several non-cognate amino acids, including Aze. That manuscript should be cited. However, the presented data do not provide insight into whether the transfer step in ProRS is discriminative. From the relative level of charged tRNA at the 20 min time point (Fig 3d and 3e) one cannot conclude anything about rates of charging/transfer. So, the transfer step should be isolated and followed, as well as the rate of aminoacylation. Mechanistic work on both systems is required to provide insights into the steps that strengthen or weaken the specificity of the synthetic and editing reactions against non-proteinogenic amino acids. For example, is the transfer step in both cases significantly slower than the transfer of cognate and near-cognate amino acids? Is this the important discrimination step that allows pre-transfer editing of Aze in the case of HsAlaRS but not ProRS because the latter enzyme lacks that capability? Is the post-transfer editing site more specific than the synthetic site (apparently, neither enzyme hydrolyzes Aze-tRNA)?

The obvious concern when the reactions with non-cognate and cognate amino acids are kinetically indistinguishable (and the label is on tRNA) is that there is a contamination with traces of the cognate substrate in the non-cognate amino acid sample. Was Aza checked for the presence of Pro? The authors did not comment on that. For example, lack of Aze editing by ProXP-ala factor (Fig 3c) may occur because the substrate used was actually Pro-tRNA^{Pro}. Indeed, during mischarging of tRNA^{Pro} with Aza, Pro-tRNA^{Pro} could be formed with the Pro contaminants if they exist in the Aza sample (mischarging was performed with 500 mM Aza; if Pro is present at 0.1% it will give a concentration of proline (5xK_M) that is substantially above the concentration of tRNA used). Such levels of cognate amino acid contamination have been observed in other systems. It is clear from MS analysis that Aze is misincorporated into proteins. However, what was the mistranslation level and how does this correlate with in vitro data? Is it possible that some effects of Aze are masked with Pro, like some (weak) editing by ProXP-ala? Also, the competition assay and the measured tRNA charging levels at the long (20 min) time point (Fig 3d and 3e) are relatively insensitive to the contamination issue (in the experiments presented at Fig3d if 0.1% of contamination is present the amount of Pro is similar to the amount of tRNA, so at longer time points high levels of charging might be observed – reaction rates are needed). The authors may want to address and clarify this issue.

In vivo data and MS analysis were performed under conditions that favor the incorporation of Aze at Pro sites (1 mM – 5mM concentration of Aze is 1/3 – 2 x K_M (Aze) for ProRS but 1/40 – 1/8 x K_M (Aze) for Hs AlaRS). Although I agree that a concentration of 1 mM could be physiologically relevant (is it known what concentration of Aze can be found in humans after food ingestion?), it may be worth seeing whether at higher concentrations of Aze some misincorporation occurs at Ala sites (I am referring to the observed 25% level of charging with FL AlaRS). The mistranslation levels should then be compared with the levels of Aza incorporation at Pro sites. Although it presumably does not present a physiologically relevant situation, it will reveal whether discrimination against Aze is mostly based on the K_M for the activation reaction, as it is substantially higher for AlaRS (38.5 mM) than for ProRS (3 mM).

Specific comments

Main text-

lines 37 and 38. Pre-transfer editing should be mentioned here as well.

line 75, please provide MS spectra and level of mistranslation.

lines 88-92 Please comment that pre-transfer editing occurs in the synthetic active site.

line 107 Please cite your work (Beuning and Musier-Forsyth, 2001, J. Biol. Chem. 276, 30779–30785) in which you demonstrated that Hs ProRS lacks the pre-transfer editing activity against Aze

line 108 – using the term “editing site” in terms of pre-transfer editing in AlaRS is confusing; the term is generally used for post-transfer editing. So please rephrase.

line 109 – “it evaded the other (ProRS)” do you mean pre-transfer editing of ProRS? You published earlier that human ProRS does not exhibit pre-transfer editing against several non-cognate amino acids, including Aze. So you need the citation here as well. Or do you mean ProXp-ala?

Fig 3 - How was the level of charged tRNA normalized (Fig 3b) and what was the maximal charging level of tRNA used (Figs 3b and 3d)? Please comment in the manuscript

Reviewer #3 (Remarks to the Author):

Song et al. For Nature Comm 2017

The misincorporation of unnatural amino acids into human proteins is a poorly understood health hazard. The toxicity caused by such residues has been poorly characterized, and the mechanisms that prevent their incorporation (or fail to do so) are only partially understood.

In this manuscript, Song et al. characterize the process of Aze misincorporation to the human proteome. They demonstrate that this amino acid can be initially bound by AlaRS and ProRS.

Interestingly, only AlaRS manages to prevent Aze misincorporation through a pre-transfer editing mechanism. ProRS, on the other hand, generates Aze-tRNA_{Pro}, generating random Pro to Aze mutations in proteins.

The authors also show that Aze exposure is toxic to HeLa cells, an effect reversed by competition with Pro. It remains to be seen what is the reason for Aze toxicity in human cells, however. Translation is not the only pathway that uses proline, and azetidine could be interfering with other aspects of metabolism.

In this regard the authors should submit a quantified analysis of the degree of azetidine misincorporation observed in cultured cells. This is far from being physiological, but at least it would provide readers with an indication of the extent of Aze incorporation, and it is a useful measure of the toxic potential of the residue.

Also, a discussion as to the reasons why AlaRS can discriminate Aze but ProRS can't would be important. Is there a structural reason why this may be so ?

In all other regards this is an extremely clear manuscript that contributes very useful information to the issue of non-natural amino acid toxicity.

Responses to Reviewers' Comments:

Reviewers' comments:

Reviewer #1 (Remarks to the Author):

In this manuscript, the authors report that Aze is confused by AlaRS and ProRS for Ala or Pro both. Also it is reported that Aze is misincorporated into protein as Pro on the basis that AlaRS can edit Aze-tRNA(Ala), whereas ProRS fails to edit Aze-tRNA(Pro). This is very interesting paper and a significant body of work, which is suitable for publication in Nature Communications. But there are some concerns need to be addressed.

1. In Fig. 1, omit electron density maps of Aze are not presented.

Omit electron density maps are shown in figure 1c and e.

2. In Fig. 2f, the activation of Aze by AlaRS by is too slight—slighter than the misactivation of Ser. To clearly indicate that AlaRS activates Aze, the data AlaRS does not activate some other amino acid is required as a negative control. Otherwise the description “both enzymes activated Aze (page 4, line 81)” should be toned down.

A figure with negative controls has been added (figure 4c).

3. In Fig. 3d, negative controls for competition experiments should be supplied.

Serine was used as a negative control for competition with proline for tRNA^{Pro} charging by Hs ProRS.

Data are shown in Supplementary figure 3b.

4. Please mark the labels of AlaRS domains in Fig. 3b.

Manuscript updated.

Reviewer #2 (Remarks to the Author):

The manuscript by Song et al demonstrates that azetidine-2-carboxylic acid, a non-proteinogenic amino acid found in some vegetables, acts as a proline and alanine analogue in the activation/aminoacylation reactions of human ProRS and AlaRS, respectively. At the same time, Aze evades the proofreading pathways that control the accuracy of proline translation, but is edited by the AlaRS pre-transfer editing activity. Misincorporation of Aze at proline sites was demonstrated in the case of human myelin basic protein expressed in HEK 293 cells. The observed mistranslation potentially explains the cellular toxicity of Aze that was demonstrated on both HeLa cells and zebrafish embryos.

The manuscript presents interesting findings regarding the toxicity of non-proteinogenic amino acids that are able to evade AARS quality control and participate in protein synthesis. These amino acids are present in food and were proposed to exert harmful effects in humans. Understanding the mechanisms by which they act and by which cell can defend itself from their toxicity is important. However, the manuscript lacks insights into the mechanisms underlying Aze participation in the AARS synthetic and editing pathways. Thus in this form, it does not substantially contribute to novel understanding in the field of translational quality control.

In particular, the authors showed that Aze can be activated and edited at the pre-transfer editing level by Hs AlaRS. They solved the crystal structure of AlaRS:Aze-AMS, and report that it strongly resembles the structure of AlaRS bound to the cognate Ala-AMS. However, as Aze-AMP is hydrolyzed within the Hs AlaRS active site (due to pre-transfer editing of Aze), one would expect to observe some differences between these two structures. Did the authors try to find a putative catalytic water? Could they provide

any structural insights into why Aze is edited in the catalytic active site of AlaRS and Ala is not? And if not, can they comment on that?

Although the reviewer's question is a good one, because crystal structures only capture a single state of the protein conformation, we were not able to obtain more in depth information from the structures on the mechanism of pre-transfer editing. In further inspection of the structures, we did not find any catalytic water in the structure, and have pointed this out in the revised manuscript.

The data showed that pre-transfer editing does not contribute to the editing of serine during Ser-tRNA^{Ala} synthesis (Fig 3b; see the full aminoacylation profile with AD), while in the case of Aze it seems to be responsible for the low level of Aze-tRNA^{Ala} formation.

Thank you. The editing-domain deletion we used (to create the AD) ablates the C-Ala domain at the very C-terminus of AlaRS, and that domain contributes to the tRNA binding efficiency. Further, early work of Jasin et al. (Jasin et al., Cell, 1984 and Jasin et al., JBC, 1985) showed that, at least with the well conserved E. coli AlaRS, mutations in the region that was ablated do affect the catalytic site, presumably through some sort of communication with the active site. We believe, therefore, that the incremental effect of the "editing domain" on charging activity (through, but not only, the C-Ala segment) probably explains the paradox. Because the differences in the referred-to figure (now figure 5b) are not large, we respectfully suggest it is not worth further pursuing.

Meanwhile, the pre-transfer editing rates for both serine and Aze are highly similar and (presumably) slow (Fig 3a, can you please provide the rates). Why do similarly slow tRNA-independent pre-transfer editing reactions contribute to the editing only in the case of Aze, and not Ser? This should be explained and further experimentally addressed by measuring the isolated transfer step (observed mischarging up to 25% indicates it may be possible) and other editing assays. The question is important as the data suggest participation of the transfer step in discrimination against Aze.

This experiment is best done (as performed in this laboratory in the past) by first isolating the Aze-radio-labeled bound adenylate and then measuring transfer to the separately added tRNA. For this purpose, we are not aware of any source for radioactive Aze. Here again, the experiment is certainly something to

consider for investigators wishing to pursue the issue in more detail, but we respectfully submit it is well beyond the spirit and intent of what we were trying to establish in this work.

The authors showed that Aze mimics Pro in Pro-tRNA^{Pro} synthesis and escaped the editing function of Hs ProRS (Fig 3d and 3e). This is in agreement with their previous work (Beuning and Musier-Forsyth, 2001, J. Biol. Chem. 276, 30779–30785) where they demonstrated that Hs ProRS lacks the pre-transfer editing activity against several non-cognate amino acids, including Aze. That manuscript should be cited.

However, the presented data do not provide insight into whether the transfer step in ProRS is discriminative. From the relative level of charged tRNA at the 20 min time point (Fig 3d and 3e) one cannot conclude anything about rates of charging/transfer. So, the transfer step should be isolated and followed, as well as the rate of aminoacylation. Mechanistic work on both systems is required to provide insights into the steps that strengthen or weaken the specificity of the synthetic and editing reactions against non-proteinogenic amino acids. For example, is the transfer step in both cases significantly slower than the transfer of cognate and near-cognate amino acids? Is this the important discrimination step that allows pre-transfer editing of Aze in the case of HsAlaRS but not ProRS because the latter enzyme lacks that capability? Is the post-transfer editing site more specific than the synthetic site (apparently, neither enzyme hydrolyzes Aze-tRNA)?

We have now cited the Beuning and Musier-Forsyth paper. Thank you for catching our omission.

As stated, we do not have the reagent needed to measure the transfer step and feel the experiment in any case is not within the scope of this work.

However, we do provide the time course of Aze competition with Pro for Hs ProRS charging tRNA^{Pro} (Supplementary figure 3a) and the time course for Hs ProRS charging of tRNA^{Pro} (Supplementary figure 3c).

In any case, the essential point is that Hs AlaRS was able clear away Aze by pre-transfer editing, while both post-editing sites for Hs AlaRS and Hs ProRS were unable to correct errors caused by Aze.

The obvious concern when the reactions with non-cognate and cognate amino acids are kinetically indistinguishable (and the label is on tRNA) is that there is a contamination with traces of the cognate

substrate in the non-cognate amino acid sample. Was Aza checked for the presence of Pro? The authors did not comment on that. For example, lack of Aze editing by ProXP-ala factor (Fig 3c) may occur because the substrate used was actually Pro-tRNA^{Pro}. Indeed, during mischarging of tRNA^{Pro} with Aza, Pro-tRNA^{Pro} could be formed with the Pro contaminants if they exist in the Aza sample (mischarging was performed with 500 mM Aza; if Pro is present at 0.1% it will give a concentration of proline (5xKM) that is substantially above the concentration of tRNA used). Such levels of cognate amino acid contamination have been observed in other systems.

Aze stock was checked for amino acid contamination. No amino acid contamination above 0.01% was detected.

It is clear from MS analysis that Aze is misincorporated into proteins. However, what was the mistranslation level and how does this correlate with in vitro data? Is it possible that some effects of Aze are masked with Pro, like some (weak) editing by ProXP-ala? Also, the competition assay and the measured tRNA charging levels at the long (20 min) time point (Fig 3d and 3e) are relatively insensitive to the contamination issue (in the experiments presented at Fig3d if 0.1% of contamination is present the amount of Pro is similar to the amount of tRNA, so at longer time points high levels of charging might be observed – reaction rates are needed). The authors may want to address and clarify this issue.

Same as above.

In vivo data and MS analysis were performed under conditions that favor the incorporation of Aze at Pro sites (1 mM – 5mM concentration of Aze is $1/3 - 2 \times \text{KM (Aze)}$ for ProRS but $1/40 - 1/8 \times \text{KM (Aze)}$ for Hs AlaRS). Although I agree that a concentration of 1 mM could be physiologically relevant (is it known what concentration of Aze can be found in humans after food ingestion?), it may be worth seeing whether at higher concentrations of Aze some misincorporation occurs at Ala sites (I am referring to the observed 25% level of charging with FL AlaRS). The mistranslation levels should then be compared with the levels of Aza incorporation at Pro sites. Although it presumably does not present a physiologically relevant situation, it will reveal whether discrimination against Aze is mostly based on the Km for the activation reaction, as it is substantially higher for AlaRS (38.5 mM) than for ProRS (3 mM).

The concentration of Aze found in humans after food ingestion has not been determined, and should vary widely based on the amounts of beet consumed. At high concentrations of Aze (40 mM), we found that most cells are dead within 24 hrs (Supplementary figure 1d). 5 mM Pro was able to rescue most of the cell death, while 5 mM Ala alone could not rescue cell death. However when we added Ala in the presence of Pro, the viable cell number slightly increased, compared to the Pro treated cells, showing that at high concentrations Ala may be inserted into proteins (Supplementary figure 1d). Because most cells die at 40 mM Aze, we could not determine the mistranslation levels at 40 mM Aze.

Specific comments

Main text-

lines 37 and 38. Pre-transfer editing should be mentioned here as well.

We made the change.

line 75, please provide MS spectra and level of mistranslation.

We made the change.

lines 88-92 Please comment that pre-transfer editing occurs in the synthetic active site.

We made the change.

line 107 Please cite your work (Beuning and Musier-Forsyth, 2001, J. Biol. Chem. 276, 30779–30785) in which you demonstrated that Hs ProRS lacks the pre-transfer editing activity against Aze.

We made the change.

line 108 – using the term “editing site” in terms of pre-transfer editing in AlaRS is confusing; the term is generally used for post-transfer editing. So please rephrase.

We made the change.

line 109 – “it evaded the other (ProRS)” do you mean pre-transfer editing of ProRS? You published earlier that human ProRS does not exhibit pre-transfer editing against several non-cognate amino acids, including Aze. So you need the citation here as well. Or do you mean ProXp-ala?

We made the change.

Fig 3 - How was the level of charged tRNA normalized (Fig 3b) and what was the maximal charging level of tRNA used (Figs 3b and 3d)? Please comment in the manuscript.

For Figure 3b, the level of charged tRNA was normalized to the Ala-tRNA(Ala) aminoacylated by the FL protein. The maximal charging level of Ala-tRNA(Ala) was about 40%. Before normalization, all of the levels of charged tRNA were corrected by subtracting the background in the absence of amino acid with the corresponding protein. For Figure 3d, approximately 50% of tRNA^{Pro} was charged at maximal level.

Reviewer #3 (Remarks to the Author):

Song et al. For Nature Comm 2017

The misincorporation of unnatural amino acids into human proteins is a poorly understood health hazard. The toxicity caused by such residues has been poorly characterized, and the mechanisms that prevent their incorporation (or fail to do so) are only partially understood.

In this manuscript, Song et al. characterize the process of Aze misincorporation to the human proteome.

They demonstrate that this amino acid can be initially bound by AlaRS and ProRS.

Interestingly, only AlaRS manages to prevent Aze misincorporation through a pre-transfer editing mechanism. ProRS, on the other hand, generates Aze-tRNA_{Pro}, generating random Pro to Aze mutations in proteins.

The authors also show that Aze exposure is toxic to HeLa cells, an effect reversed by competition with Pro. It remains to be seen what is the reason for Aze toxicity in human cells, however. Translation is not the only pathway that uses proline, and azetidine could be interfering with other aspects of metabolism.

In this regard the authors should submit a quantified analysis of the degree of azetidine misincorporation observed in cultured cells. This is far from being physiological, but at least it would provide readers with an indication of the extent of Aze incorporation, and it is a useful measure of the toxic potential of the residue.

We updated our mass spec data with increased coverage and more in-depth analysis. To increase the

mammalian cell purified-protein yield and spectral counts, we used the maltose-binding protein (MBP). New data are shown in figure 3 and tables 1 & 2. We obtained 96% sequence coverage, and of the 21 Pro positions, they were collectively replaced at a frequency of 13.8% with Aze. In our extended analyses we also found that one Ala position was replaced by Aze at a frequency of 0.024%. This strengthened our argument that Aze is activated by Hs AlaRS, and with rare exception is cleared away by editing.

Also, a discussion as to the reasons why AlaRS can discriminate Aze but ProRS can't would be important.

Is there a structural reason why this may be so?

We were not able to resolve any differences that may explain why AlaRS discriminates against Aze, but ProRS cannot. For example, we could not find an active-site bound water in one structure and not the other. However, while Hs AlaRS has pretransfer activity, Hs ProRS has only a weak pre-transfer editing activity (Beuning et al., JBC, 2001). This difference correlates with our observations and perhaps is relevant.

In all other regards this is an extremely clear manuscript that contributes very useful information to the issue of non-natural amino acid toxicity.

Reviewers' Comments:

Reviewer #1 (Remarks to the Author):

I am satisfied with the addresses by the authors, and recommend publication of this paper in Nat. Commun.

Reviewer #2 (Remarks to the Author):

The transfer step can be successfully measured using [32P] labeled tRNA (a pre-formed AARS:aminoacyl-AMP complex is mixed with a limiting amount of [32P]tRNA). These experiments could address very interesting mechanistic aspects of the nonproteinogenic amino acid discrimination by AARSs. However, I agree they are slightly out of the scope of the manuscript. The authors have addressed my other concerns.

Reviewer #3 (Remarks to the Author):

The authors have addressed my concerns very satisfactorily. I find the MS data convincing, and it significantly strengthens the conclusions.